# Effects of Copper Surface Oxidation and Reduction on Shear-Bond Strength Using Functional Monomers

**DOI:** 10.3390/ma14071753

**Published:** 2021-04-02

**Authors:** Haruto Hiraba, Hiroyasu Koizumi, Akihisa Kodaira, Kosuke Takehana, Takayuki Yoneyama, Hideo Matsumura

**Affiliations:** 1Department of Fixed Prosthodontics, Nihon University School of Dentistry, 1-8-13, Kanda-Surugadai, Chiyoda-ku, Tokyo 101-8310, Japan; kodaira.akihisa@nihon-u.ac.jp (A.K.); deko18027@g.nihon-u.ac.jp (K.T.); matsumura.hideo@nihon-u.ac.jp (H.M.); 2Division of Advanced Dental Treatment, Dental Research Center, Nihon University School of Dentistry, 1-8-13, Kanda-Surugadai, Chiyoda-ku, Tokyo 101-8310, Japan; 3Department of Dental Materials, Nihon University School of Dentistry, 1-8-13, Kanda-Surugadai, Chiyoda-ku, Tokyo 101-8310, Japan; koizumi.hiroyasu@nihon-u.ac.jp (H.K.); yoneyama.takayuki@nihon-u.ac.jp (T.Y.); 4Division of Biomaterials Science, Dental Research Center, Nihon University School of Dentistry, 1-8-13 Kanda-Surugadai, Chiyoda-ku, Tokyo 101-8310, Japan; 5Applied Oral Sciences & Community Dental Care, Faculty of Dentistry, The University of Hong Kong, Hong Kong SAR 999077, China

**Keywords:** acrylic resin, adhesive, dental metal, functional monomer, oxide film

## Abstract

This study was conducted to clarify the influence of the copper surface oxidation and reduction on the shear-bond strength with functional monomers. Unheated copper specimens (UH; *n* = 88) were wet-ground. Three-quarters of the UH were then heated (HT). Two-thirds of the HT was then immersed in a hydrochloric acid solution (AC). Half of the AC was then reheated (RH). Each group was further divided into two groups (*n* = 11), which were primed by either 6-methacryloyloxyhexyl 2-thiouracil-5-carboxylate (MTU-6) or 10-methacryloyloxydecyl dihydrogen phosphate (MDP). The shear-bond strength tests were used for bonding with an acrylic resin. The surface roughness values and chemical states of the four groups were analyzed using a confocal scanning laser microscope and X-ray photoelectron spectroscopy (XPS). The shear-bond strengths of HT and RH were the lowest in the MTU-6-primed groups. The result of AC was significantly lower than others in the MDP-primed groups. The XPS results showed that the surfaces of UH and AC consisted of Cu_2_O and Cu. The surface changed to CuO upon heating. The presence or absence of copper-oxide films showed the opposite trends in the effectiveness of MTU-6 and MDP to improve bond strength. The results could elucidate the effects of functional monomers on copper-oxide films.

## 1. Introduction

The combination of constituent metal elements imparts noble-metal alloys with corrosion resistance and mechanical strength. Many dental noble-metal alloys contain copper. Although copper and silver included in alloys are considered to be one of the causes of cytotoxicity [1,2], copper is an important component for improving the mechanical strength of alloys. Owing to their increased copper contents, type 3 and type 4 of dental casting gold alloys retain sufficient mechanical strength for application to frameworks of fixed partial dentures [3]. The copper surface changes from Cu_2_O to CuO by heating [4]. As this oxidation-state transition also occurs in copper-containing alloys or compounds [5,6], copper plays an important role in adhesive bonding.

Surface treatment with adhesive functional monomers is an effective method of bonding dental alloys with acrylic resins. Dentists choose a functional monomer depending on the kind of alloy they are using. Usually, acidic functional monomers and organic sulfur compounds are applied to non-noble metal alloys and noble-metal alloys, respectively [7]. Acidic functional monomers, e.g., carboxylic acids and phosphates, act on the oxide film on the surfaces of non-noble metals. As such monomers do not impart any advantages to noble-metal alloys that barely form surface oxide films, oxidation treatments are required. In previous studies [8,9,10], copper oxidation has improved the adhesion effectiveness of acidic functional monomers on noble-metal alloys. Copper held an important role in oxidation treatments for noble-metal alloys at the time when the mainstream functional monomers were acidic functional monomers. Organic sulfur compounds are often used for noble-metal alloys at the present moment. In contrast to acidic functional monomers, organic sulfur compounds act directly on the surfaces of noble metals [11]. The first known application of organic sulfur compounds to adhesion was a study on the corrosion resistance of copper [12]. The thione-thiol tautomer in organic sulfur compounds strongly adsorbs onto noble-metal surfaces [11,13,14]. One organic sulfur compound that has shown stable high bonding strength for noble-metal alloys is 6-methacryloyloxyhexyl 2-thiouracil-5-carboxylate (MTU-6) [9,15,16,17]. Although MTU-6 has shown stronger adsorption on the surfaces of noble-metal alloys [16] and has bonded to copper better than any other organic sulfur compound [15], copper oxidation remarkably reduces the adhesion effectiveness of MTU-6 [18].

Applying primers containing both organic sulfur compounds and acidic functional monomers has been shown to be more effective than applying only organic sulfur compounds to improve adhesion strengths of noble-metal alloys in previous studies [10,17,19,20]. The results of these studies have suggested that acidic functional monomers act on copper oxide films formed by airborne-particle abrasion on the surfaces of noble-metal alloys [10,20]. At the copper surface, Cu (Cu^0^) changes to Cu_2_O (Cu^+^) or CuO (Cu^2+^) depending on the treatments used. As these changes in the copper surface and the effects of functional monomers on the copper surface are still not completely understood, this study was conducted to clarify the difference between how MTU-6 (an organic sulfur compound) and MDP (an acidic functional monomer) change the copper surface.

## 2. Materials and Methods

### 2.1. Materials

The materials are listed in Table 1. The adherend material was copper metal (Nilaco Corp., Tokyo, Japan). Two single-liquid primers were used: Metaltite (MT; Tokuyama Dental Corp., Tokyo, Japan) containing 6-methacryloyloxyhexyl 2-thiouracil-5-carboxylate (MTU-6) in ethanol and Super-Bond PZ Primer Liquid A (PZ; Sun Medical Co., Ltd., Moriyama, Japan) containing 10-methacryloyloxydecyl dihydrogen phosphate (MDP) in methyl methacrylate (MMA). A tri-*n*-butylborane (TBB; Super-Bond C&B Catalyst V, Sun Medical Co., Ltd., Moriyama, Japan)-initiated acrylic resin (MMA-TBB resin) was used as a luting material. The powder (Super-Bond C&B Opaque Ivory Powder, Sun Medical Co., Ltd., Moriyama, Japan) of MMA-TBB resin consisted of finely pulverized poly (methyl methacrylate) (PMMA) and PMMA-coated titanium dioxide. The monomer liquid was MMA (Tokyo Chemical Industry Co., Ltd., Tokyo, Japan) without functional monomer.

### 2.2. Preparation of Specimens

Copper specimens (*n* = 88) were prepared in the same shape as in the previous report [18]. The specimens were divided into four treatment groups (*n* = 22) according to surface conditioning before bonding, as indicated in Figure 1. All the unheated (UH, Figure 2a) specimens were initially wet-ground with 1500-grit silicon carbide abrasive paper (WetorDry Sheet, 3M Corp., St. Paul, MN, USA). Three-quarters of the specimens (*n* = 66) were first heated (HT, Figure 2b). Subsequently, 44 of the heated specimens were immersed in a hydrochloric acid (HCl) solution (AC, Figure 2c). Half of the AC specimens (*n* = 22) were then reheated (RH, Figure 2d).

#### 2.2.1. Heating

The heating conditions were based on those described in previous reports [8,9,18]. The specimens were heated at 400 °C for 4 min and then were left to stand until cooling at room temperature. The specimens were subsequently reheated under the same conditions.

#### 2.2.2. Immersion in HCl Solution

The AC and RH specimens were immersed in a concentrated HCl solution (35 to 37%, Wako Pure Chemical Industries Ltd., Osaka, Japan) for 1 h and were subsequently rinsed with acetone to remove any residual HCl and then dried.

### 2.3. Surface Roughness Measurements

The surface roughness (*Sa*) values of all the samples were analyzed using a confocal scanning laser microscope (1LM21W, Lasertec Corp., Yokohama, Japan) equipped with a He–Ne laser light source operating at a wavelength of 633 nm. The microscope resolution was ~0.03 µm, the objective-lens magnification was 20×, and the focal depth in the *Z*-axis scan range was 24 µm. *Sa* was measured in five areas (400 μm × 400 μm) randomly selected from the surface of each specimen.

### 2.4. Specimen Preparation for Shear-Bond Strength Tests

The specimens of the four treatment groups were primed by MT or PZ (*n* = 11). The primed specimens were prepared for the shear-bond strength tests according to the same method described in the previous report [18]. The experimental conditions of the shear-bond strength tests were determined using a mechanical testing device (Type 5567, Instron, Canton, MA, USA) at a crosshead speed of 0.5 mm/min. The failure modes of the specimen bonded areas were classified after shear-bond strength tests.

### 2.5. Statistical Analysis

The results of the shear-bond strength tests were analyzed. The D’Agostino–Pearson omnibus test (GraphPad Prism 8, GraphPad Software, Inc., La Jolla, CA, USA) was used and confirmed a normal distribution. Furthermore, the Kruskal–Wallis (KyPlot 6.0, KyensLab, Inc., Tokyo, Japan), Steel–Dwass (KyPlot 6.0), and Mann–Whitney *U* (GraphPad Prism 8) tests were performed. The surface roughness results were analyzed by the Kruskal–Wallis (KyPlot 6.0) and Steel–Dwass (KyPlot 6.0) tests. The Steel–Dwass test was used to compare the significant differences between the four types of surface treatments. The Mann–Whitney *U* test was used to compare the significant differences between the two primers. The statistical significance was set at *α* = 0.05.

### 2.6. X-Ray Photoelectron Spectroscopy (XPS) Analysis

Before the bonding tests, the surfaces of all the specimens were chemically analyzed by X-ray photoelectron spectroscopy (XPS; ESCA-3400, Shimadzu Corp., Kyoto, Japan) with Mg-K*α* radiation.

## 3. Results

### 3.1. Surface Roughness

The surface roughness (*Sa*) values for the UH, HT, AC, and RH specimen groups are presented in Table 2. The *Sa* values for the HT and RH groups were not significantly different (*p* < 0.05). The UH and AC groups showed the highest and lowest *Sa* values, respectively. The median *Sa* values were in the range 0.52 µm–0.62 µm.

### 3.2. Shear-Bond Strength Tests

The median shear-bond strengths were in the ranges of 7.0–27.6 and 3.5–26.1 MPa for the MT- and PZ-primed specimen groups (MT and PZ groups), respectively, as shown in Table 3. AC-MDP did not result in a normal Gaussian distribution (GraphPad Prism 8) of the shear-bond strength. According to the significant differences in the shear-bond strengths (*p* < 0.05), the Steel–Dwass test revealed that the specimens could be further divided into two subgroups within the MT and PZ groups, depending on the primer used. The shear-bond strengths of the UH and AC specimens were the highest in the MT group, whereas those of the HT and RH specimens were the lowest in the same group. The shear-bond strengths of the UH, HT, and RH specimens were significantly higher than those of the AC specimens in the PZ group. The Mann–Whitney *U* test revealed that all the specimens subjected to the same surface treatments showed significantly different shear-bond strengths (*p* < 0.05) for the different primers.

The results obtained for the adherend surfaces are also summarized in Table 3. The failure mode was detected by a combination of adhesive and cohesive failures under most of the experimental conditions. All the HT-MTU-6, RH-MTU-6, and AC-MDP specimens showed adhesive failure at the acrylic resin–metal interface. None of the specimens showed only cohesive failure under any experimental conditions.

### 3.3. XPS Analysis

Changes in the copper surfaces were analyzed by XPS (Figure 3 and Figure 4). The HT and RH specimens showed more intense O 1*s* peaks than the UH and AC ones did in the wide-scan spectra (Figure 3). The UH and HT specimens showed different Cu-related peaks, such as Cu 2*p*, owing to heating; the AC and RH specimens similarly showed different Cu-related peaks. The wide-scan XPS spectra for the UH and AC specimens showed similar shapes.

The narrow-scan XPS spectra revealed the characteristics of the copper surfaces of the specimens (Figure 4). Table 4 lists the characteristic peaks. From the peak characteristics in the Cu 2*p*, Cu *LMM*, O 1*s,* and Cl 2*p* regions (Figure 4), Cu_2_O (Cu^+^) and Cu (Cu^0^) both had formed on the surface of the unheated copper specimens. In contrast, the copper surfaces of the HT and RH specimens showed peaks attributable to CuO (Cu^2+^) immediately after heating (Figure 4a–c). The peaks attributed to CuO in the Cu 2*p* region are relatively broad and are accompanied by characteristic satellite peaks. The copper surface immersed in a concentrated HCl solution (AC) showed different from peaks attributable to CuO of HT (Figure 4). Although their peak positions had shifted, the narrow-scan spectra of the AC and UH specimens showed similar shapes in the Cu 2*p* region. The peaks in the Cu *LMM* region showed that the surface of the AC specimen consisted of Cu_2_O, CuCl_2_, and Cu (Figure 4b and Table 4). The peak at 534.7 eV was attributed to the gas-phase water molecules (Figure 4c). Figure 4d shows the narrow-scan XPS spectra generated in the Cl 2*p* region. The peak at 199.2 eV, as listed in Table 4, is attributed to CuCl or CuCl_2_. The narrow-scan spectra for the AC specimens did not show any peaks (Figure 4d). Although the narrow-scan XPS spectra for the RH specimens showed broad peaks (including the ones at 199.2 eV), the narrow-scan XPS spectra in the Cu *LMM* region revealed that the AC-specimen surfaces did not contain CuCl or CuCl_2_ (Figure 4b,d).

## 4. Discussion

Noble-metal alloys containing copper are widely used in dentistry. Copper or silver has been reported to show an in vitro cytotoxic effect [1,2]; however, the occurrence of local and systemic toxic effects is not well known [27]. The effectiveness of using noble-metal alloys for frameworks of fixed partial dentures in dental practice is obvious, and obtaining a strong bond between noble-metal alloys and teeth is an important issue to obtain good long-term results. Therefore, the copper contained in the alloy was focused for a strong bond. This study compared the effects of MTU-6 (an organic sulfur compound) and MDP (an acidic functional monomer) on the shear-bond strengths of four copper surfaces. The previous study showed that copper oxidation inhibits the effects of MTU-6 [18]. Although Miyahara et al. [10] suggested that a copper-containing gold–silver–palladium alloy oxidized by airborne-particle abrasion or heating rendered MDP effective, the reason remains unclear because alloys may be influenced by other metal elements. The results of this copper-focused study could elucidate the effects of an oxide film to functional monomers and complement the previous studies on the adhesive bonding between metals and acrylic resins.

Although the surface roughness results obtained for the specimens were significantly different (*p* < 0.05), all the measured values were in the range ≤0.1 μm (Table 2). The slight differences in surface roughness among the specimens suggest that the different treatments did not affect the mechanical bonding between the acrylic resin and the specimens.

The results of the shear-bond strength tests showed striking differences between the effects of the functional monomers on the bonding strengths of the heat- and HCl-treated specimens (Table 3). The shear-bond strengths of the UH and AC specimens were higher than those of the HT and RH ones in the MTU-6-primed groups. As in the previous study, the effect of MTU-6 on the shear-bond strength of copper was remarkably reduced by heating [18]. By contrast, the shear-bond strengths of the UH, HT, and RH specimens were higher than that of the AC one in the MDP-primed groups. The high bond strength of the MDP-primed UH specimen is consistent with the corresponding result in the previous report and is identically expected to remarkably decrease after thermocycling [15]. The MTU-6-primed UH specimen showed a higher bond strength than the MDP-primed UH one. MDP effectively improved the bond strengths of the HT and RH specimens. The effects of MTU-6 and MDP on the bond strengths of the HT, AC, and RH specimens were in complete contrast. The results of the failure-mode tests were strongly correlated with those of the shear-bond strength tests.

The properties of the four copper surfaces were revealed by XPS and were compared with previous experimental findings reported elsewhere in the literature [21,22,23,24,25,26,28,29]. The unheated (UH) copper surface consisted of both Cu (Cu^0^) and Cu_2_O (Cu^+^) (Figure 4 and Table 4), gradually changing from Cu to Cu_2_O over time [4,24,26]. The adhesive-bonding study by Yamashita et al. [15] showed that the surface of thermocycled debonded copper consisted of Cu_2_O. The heated (HT) copper surface was covered with CuO (Cu^2+^), as in previous studies [18,22]. The AC specimens showed peaks attributable to Cu^2+^ in the Cu 2*p* region, whereas the characteristic satellite peaks attributable to CuO and CuCl_2_ [23,24,26,28] were not clearly observed (Figure 4a). This absence of the satellite peaks suggested that the formed copper-oxide layer (oxide film) had been removed by HCl, as previously reported for a copper-containing gold–silver–palladium alloy [10]. The narrow-scan spectra (Figure 4 and Table 4) suggested that CuCl_2_ (Cu^2+^ and Cl^−^) and the gas-phase water (H_2_O) occurred by a chemical reaction between CuO and HCl [25,26,29]. The spectra in the Cu *LMM* region indicated that the AC specimen surface was not covered by CuO, and the surface consisted of CuO_2_, CuCl_2_, and Cu (Figure 4b and Table 4). The surface of the reheated (RH) copper specimen also showed CuO. Although the narrow-scan spectra of the Cl 2*p* region did not necessarily preclude the existence of CuCl or CuCl_2_ on the specimen surface, the narrow-scan spectra for the Cu *LMM* region proved that the RH specimen surface consisted of CuO (Figure 4b,d).

The results of this study revealed that the effect of each functional monomer on shear-bond strength was closely related to the copper surface properties. MTU-6 effectively improved the bond strengths of the UH and AC specimens, whose surfaces consisted of Cu, Cu_2_O, or CuCl_2_; in contrast, MDP effectively improved those of the HT and RH specimens, whose surfaces consisted of CuO. Both MTU-6 and MDP had the higher bond strength to the unheated sample (UH) at the initial curing (24 h). However, as previously described, UH-MDP is expected to remarkably decrease after a load test. In other words, the presence or absence of a surface oxide film remarkably affected the effectiveness of MTU-6 and MDP to improve bond strength. As in the previous study [18], the oxide film inhibited the effectiveness of MTU-6. In contrast, MDP effectively acted on oxide films to improve adhesive bonding [7,10]. In that regard, the results of this study are consistent with those of the previous ones. The opposite trends observed for the effects of MTU-6 and MDP on the bond strength of the AC specimen also indicate the effect of the surface oxide film because the results of this study were obtained by changing the surfaces derived from the same copper starting material.

In recent years, the adhesion effectiveness of multipurpose primers containing multiple functional monomers has been reported [17]. The previous studies have reported that a bonding to a noble metal alloy become more effective by using an acidic functional monomer and an organic sulfur compound simultaneously, rather than only an organic sulfur compound; the reason is still not completely understood. Alumina-based airborne-particle abrasion is commonly used by dentists and is mainly intended to clean and mechanically roughen metal surfaces. Although some studies have reported that a copper-containing noble-metal dental alloy could be oxidized a copper by alumina-based airborne-particle abrasion [10,19,20], few studies have focused on the relation between changes in a noble-metal surface and the bonding mechanisms of functional monomers. The results of HT, AC, and RH in this study suggest that MDP acts on the copper oxide (CuO) surface film. This study has potential limitations. The primary limitation is that some of the noble-metal alloys used in clinical dentistry do not contain copper. Therefore, it is important to select a functional monomer whose properties match those of the metal alloy surface. Furthermore, the results described herein suggest that sulfur-containing and acidic functional monomers should be combined after alumina-based airborne-particle abrasion when bonding copper-containing noble-metal alloys with an acrylic resin.

## 5. Conclusions

Within the limitations of this study, the following conclusions can be drawn:The unheated copper surface consisted of both Cu and Cu_2_O because of a gradual change from Cu to Cu_2_O over time. A copper oxide film (CuO) was formed on the copper surface upon heating and reheating.The copper oxide surface significantly increased the effectiveness of MDP to shear-bond strengths, while MTU-6 was significantly reduced. Cu, Cu_2_O, and CuCl_2_ was formed on the surface of the copper immersed in a concentrated HCl solution as the oxide film eliminated, remarkably increasing and decreasing the effectiveness of MTU-6 and MDP, respectively, to improve shear-bond strength.The results of this study clearly showed that the presence or absence of the oxide film on the metal surface has a strong effect on shear-bond strength using functional monomers.A combination with sulfur-containing and acidic functional monomers is assumed to be effective for the noble metal alloys that contain both non-oxidizable noble metals, e.g., Au and Pt, and a copper oxide.The present results could complement the previous studies on the adhesive bonding between metals and acrylic resin and suggest that sulfur-containing and acidic functional monomers were better to combine after alumina-based airborne-particle abrasion when bonding copper-containing noble-metal alloys with an acrylic resin.The improvement of the bonding strength of noble-metal alloys by a simple method is important in the clinical setting because dentists can make the choice to apply a resin-bonded fixed partial denture to make miniaturization of the cut amount of abutment teeth of patients.

## Figures and Tables

**Figure 1 materials-14-01753-f001:**
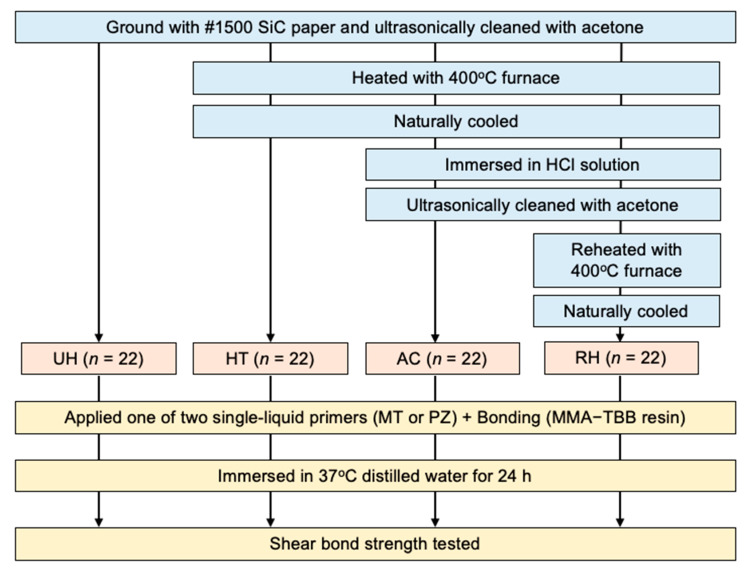
Flow diagram for preparing bonded specimens for shear-bond strength testing.

**Figure 2 materials-14-01753-f002:**
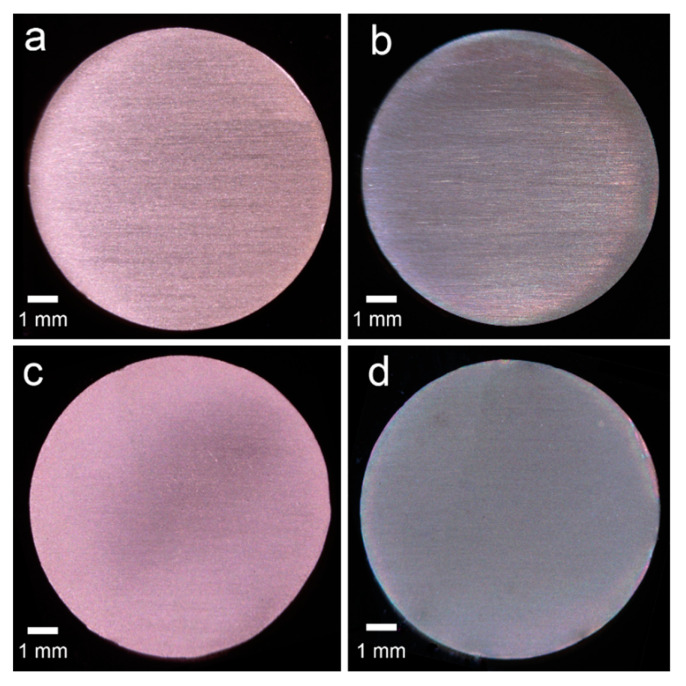
Surface-treated copper specimens. (**a**) UH: unheated; (**b**) HT: heated; (**c**) AC: heated and immersed in HCl; and (**d**) RH: heated, immersed in HCl, and reheated. Digital photographic system: original magnification 8×.

**Figure 3 materials-14-01753-f003:**
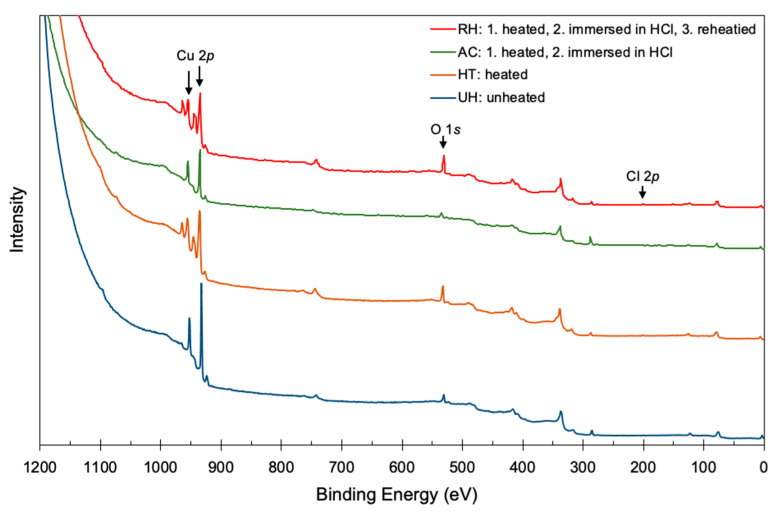
XPS wide-scan spectra of Cu plates: unheated (UH); heated (HT); heated and immersed in HCl (AC); and heated, immersed in HCl, and reheated (RH).

**Figure 4 materials-14-01753-f004:**
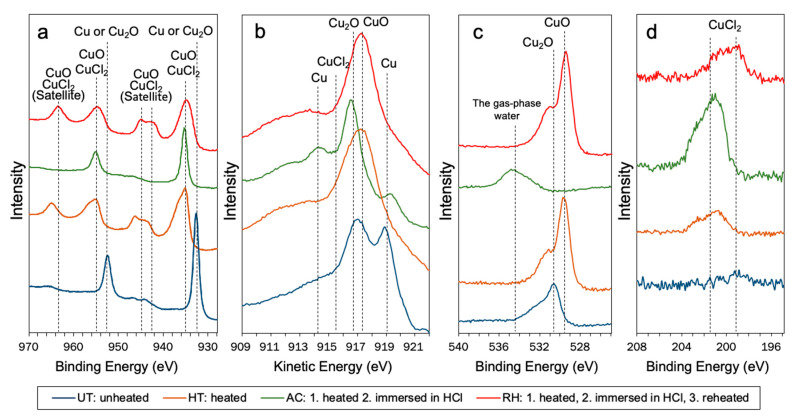
XPS narrow-scan spectra of (**a**) Cu 2*p* region, (**b**) Cu *LMM*, (**c**) O *1s* region, and (**d**) Cl 2*p* region of Cu plates.

**Table 1 materials-14-01753-t001:** Materials assessed.

Material/Trade Name	Manufacturer	Lot	Composition
Element metal	-	-	-
Copper metal	Nilaco Corp., Tokyo, Japan	44225602	Cu 99.9, mass%
Primer	-	-	-
Metaltite	Tokuyama Dental Corp., Tokyo, Japan	0382	MTU-6, Ethanol
Super-Bond PZ Primer (Liquid A)	Sun Medical Co., Ltd., Moriyama, Japan	SM1	MDP, MMA
Luting material	-	-	-
Super-Bond C&B Catalyst V	Sun Medical Co., Ltd.	SX11	TBB, TBB-O, hydrocarbon
Super-Bond C&B Opaque Ivory Powder	Sun Medical Co., Ltd.	RM1	PMMA, titanium oxide
Methyl methacrylate	Tokyo Chemical Industry Co., Ltd., Tokyo, Japan	ZJ3WJIJ	MMA, 99.8%

MTU-6: 6-methacryloyloxyhexyl 2-thiouracil-5-carboxylate; MDP: 10-methacryloyloxydecyl dihydrogen phosphate; MMA: methyl methacrylate; TBB: tri-*n*-butylborane; TBB-O: partially oxidized tri-*n*-butylborane; PMMA: poly (methyl methacrylate). The materials assessed were mostly based on the previous report [18].

**Table 2 materials-14-01753-t002:** Surface roughness *Sa* (μm).

Treatment	Median	IQR
UH	0.62 ^a^	0.006
HT	0.55 ^b^	0.009
AC	0.52 ^c^	0.005
RH	0.54 ^b^	0.006

*n* = 5; Different letters of Median indicate that the values are significantly different (Steel–Dwass test: *p* < 0.05); IQR, interquartile range; Statistical category.

**Table 3 materials-14-01753-t003:** Shear-bond strength (MPa) and failure modes after testing.

Treatment	MTU-6 (MT Group)				MDP (PZ Group)				M-U	*p*
	Median	IQR	A	CA	Median	IQR	A	CA		
UH	27.6 ^a^	2.0	0	11	25.8 ^c^	1.4	0	11	S	0.038
HT	10.1 ^b^	4.3	11	0	26.1 ^c^	3.1	0	11	S	<0.0001
AC	27.3 ^a^	2.6	0	11	3.5 ^d^	0.7	11	0	S	<0.0001
RH	7.0 ^b^	2.0	11	0	23.5 ^e^	3.1	0	11	S	<0.0001

*n* = 11; SD: Standard deviation; IQR: Interquartile range. Categories identified by different letters of Midian showed significant differences for same primer (Steel–Dwass test; *p* < 0.05) M-U: Abbreviation “S” indicates that the difference between MTU-6 and MDP for the same surface treatments is significant (Mann–Whitney *U* test; *p* < 0.05). *p*: *p* value of Mann–Whitney *U* test; A: adhesive failure at acrylic resin–metal interface; CA: combination of adhesive and cohesive failures.

**Table 4 materials-14-01753-t004:** Peaks of bonding-energy (eV) and Auger transition kinetic energies (eV).

Element	Peak Energy (eV)	Peak Assignment (Compound)	Reference
Cu 2*p3/2*	932.6	Cu_2_O or Cu	[21,22,23]
Cu 2*p1/2*	952.2	Cu_2_O or Cu	[21,22,23]
Cu 2*p3/2*	935.1	CuO	[21,22,23]
Cu 2*p3/2*	934.9	CuCl_2_	[21,22,23]
Cu 2*p3/2*	943.4, 943.6	CuO, CuCl_2_ satellite peaks	[21,22,23]
Cu 2*p1/2*	955.3, 954.7	CuO	[21,22,23]
Cu 2*p1/2*	964.9, 963.6	CuO, CuCl_2_ satellite peaks	[21,22,23]
Cu *LMM*	915.4	CuCl_2_	[21,22,23]
Cu *LMM*	916.6	Cu_2_O	[23,24]
Cu *LMM*	917.5	CuO	[23,24]
Cu *LMM*	914.3, 918.5	Cu	[23,24]
O 1*s*	529.7	CuO	[23,24]
O 1*s*	530.6	Cu_2_O	[23,24]
O 1*s*	534.7	The gas-phase water	[25]
Cl 2*p3/2*	199.2, 201.6	CuCl_2_	[23,24,26]

Cu *LMM*: peak positions expressed in kinetic energy.

## Data Availability

Not applicable.

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
