# Peer review of "Effects of Copper Surface Oxidation and Reduction on Shear-Bond Strength Using Functional Monomers"

_materials, 2021, doi:10.3390/ma14071753_

Round 1

Reviewer 1 Report

The work has been well presented and structured. I do recommend the publication of the paper after however some recommendations:

- Conclusions must be improved : For instance, the very first sentence is not understandable: " The following conclusions can be drawn from the results of this study: the polished 290 copper surface consisted of both Cu and Cu2O."

Reviewer 2 Report

Overall comments:

The subject matter of this manuscript deals the influence of the copper surface oxidation and reduction on the shear-bond strength with functional monomers.

The present study is well worth investigating and the manuscript itself is considered to be theoretically and structurally reasonable. The authors did well in stating what the goal of the paper is and did nicely with all the sections. However, the authors should more elaborate upon what is of paramount importance and significance. I worry that this masks what appears to be very important subject.

Together with this critical point, there are some specific concerns that should be addressed in a point-by-point manner. If such issues were all cleared by the authors, this paper can be re-evaluated to secure its publication.

Major concerns:

1) Materials and Methods: Why the surface roughness was assessed by confocal microscopy, not AFM? AFM is generally used due to its higher resolution and physical measurement method, which make it more reliable than optical imaging-based measurement.

2) In Figure 2, why are the images deployed as round shape? And please add the scale bars.

3) The x-axis of Figure 4b is represented as ‘kinetic energy’, while the others are marked as ‘binding energy’. Is there any reason?

Reviewer 3 Report

The article “Effects of Copper Surface Oxidation and Reduction on Shear-Bond Strength using Functional Monomers” has merits and falls within the journal's scope.

The study has interesting novel information related to clarify the influence of the copper surface oxidation and reduction on the shear-bond strength with functional monomers.

The main strength is the interesting approach to a topic that involve materials and their impact in clinical applications. In fact, the authors have obtained results that could elucidate the effects of functional monomers on copper-oxide films.

The introduction should improve some key-concepts:

  1. The authors must discuss on mechanical impact of biomaterials/TCPs on cells growth - (Please, see and also discuss “Marrelli M, Codispoti B, Shelton RM, Scheven BA, Cooper PR, Tatullo M, Paduano F. Dental Pulp Stem Cell Mechanoresponsiveness: Effects of Mechanical Stimuli on Dental Pulp Stem Cell Behavior. Front Physiol. 2018 Nov 26;9:1685.”)
  2. The conclusions should be improved with clear take-home messages and including limitations of the study.

Minor suggestions:

Authors should also consider and briefly discuss the clinical impact of their study on biological tissues, mainly on some local MSCs  that can biologically act as the immunomodulatory and pro-regenerative activities in the local environment (Please, see and discuss:  Spagnuolo, G.; Codispoti, B.; Marrelli, M.; Rengo, C.; Rengo, S.; Tatullo, M. Commitment of Oral-Derived Stem Cells in Dental and Maxillofacial Applications. Dent J (Basel) 2018, 6(4), 72. - AND -            Ballini A, Boccaccio A, Saini R, Van Pham P, Tatullo M. Dental-Derived Stem Cells and Their Secretome and Interactions with Bioscaffolds/Biomaterials in Regenerative Medicine: From the In Vitro Research to Translational Applications. Stem Cells Int. 2017;2017:6975251.).

Author may improve the discussion on the role of tissue composition and its impact on local environment.

Reviewer 4 Report

The interest of the work is clear, although its scope is a little limited, nothing more than two monomers and two treatments are examined, and the conclusions are therefore limited although they are concordant with bibliography.

Round 2

Reviewer 2 Report

The authors sincerely provided replies to some critical issues raised in the previous review stage.

It is considered that the present version of the manuscript was sufficiently well revised according to the reviewers' comments.

Thus, this manuscript would be acceptable, unless otherwise explained.

Author Response

Response to Reviewer 2 Comments

Comment: The authors sincerely provided replies to some critical issues raised in the previous review stage.

It is considered that the present version of the manuscript was sufficiently well revised according to the reviewers' comments.

Thus, this manuscript would be acceptable, unless otherwise explained.

Response: Thank you very much for providing important comments. We are thankful for the time and energy you expended.

Reviewer 3 Report

Despite authors have somewhat improved their paper, it still lacks a more multidisciplinary approach. The introduction is still poor, specifically on the clinical and biological point of view. This reviewer suggests to further improve introduction and discussion on the basis of the previous suggestions and with the aim to further increase the impact on readers. 
